# LLM Dataset Inference: Did you train on my dataset?

**Pratyush Maini**[*1,2] **Hengrui Jia**[*3,4] **Nicolas Papernot**[3,4] **Adam Dziedzic**[5]

[1]Carnegie Mellon University    [2]DatologyAI    [3]University of Toronto
[4]Vector Institute    [5]CISPA Helmholtz Center for Information Security

## Abstract

The proliferation of large language models (LLMs) in the real world has come with a rise in copyright cases against companies for training their models on *unlicensed data* from the internet. Recent works have presented methods to identify if individual text sequences were members of the model's training data, known as membership inference attacks (MIAs). We demonstrate that the apparent success of these MIAs is confounded by selecting non-members (text sequences not used for training) belonging to a different distribution from the members (e.g., temporally shifted recent Wikipedia articles compared with ones used to train the model). This distribution shift makes membership inference appear successful. However, most MIA methods perform no better than random guessing when discriminating between members and non-members from the same distribution (e.g., in this case, the same period of time). Even when MIAs work, we find that different MIAs succeed at inferring membership of samples from different distributions. Instead, we propose a new *dataset inference* method to accurately identify the datasets used to train large language models. This paradigm sits realistically in the modern-day copyright landscape, where authors claim that an LLM is trained over multiple documents (such as a book) written by them, rather than one particular paragraph. While dataset inference shares many of the challenges of membership inference, we solve it by selectively combining the MIAs that provide positive signal for a given distribution, and aggregating them to perform a statistical test on a given dataset. Our approach successfully distinguishes the train and test sets of different subsets of the Pile with statistically significant p-values $< 0.1$, without any false positives.

## 1 Introduction

Training of large language models (LLMs) on large scrapes of the web [Gem, Ope] has recently raised significant privacy concerns [Rahman and Santacana, 2023, Wu et al., 2023]. The inclusion of personally identifiable information (PII) and copyrighted material in the training corpora has led to legal challenges, notably the lawsuit between *The New York Times* and OpenAI [Gry, 2023], among others [Bak, 2023, Sil, 2023]. Such cases highlight the issue of using copyrighted content without attribution and/or license. Potentially, they undermine the rights of creators and disincentivize future artistic endeavors due to the lack of monetary compensation for works freely accessible online. This backdrop sets the stage for the technical challenge of identifying training data within machine learning models [Maini et al., 2021, Shokri et al., 2017]. Despite legal ambiguities, the task holds critical importance for understanding LLMs' operations and ensuring data accountability.

Membership inference [Shokri et al., 2017] is a long-studied privacy problem, intending to infer if a given data point was included in the training data of a model. However, identifying example membership is a challenging task even for models trained on small datasets Carlini et al. [2022], Duan et al. [2023a,b], and Maini et al. [2021] presented an impossibility result suggesting that as the size of

---

*Equal contribution. Code is available at https://github.com/pratyushmaini/llm_dataset_inference/.

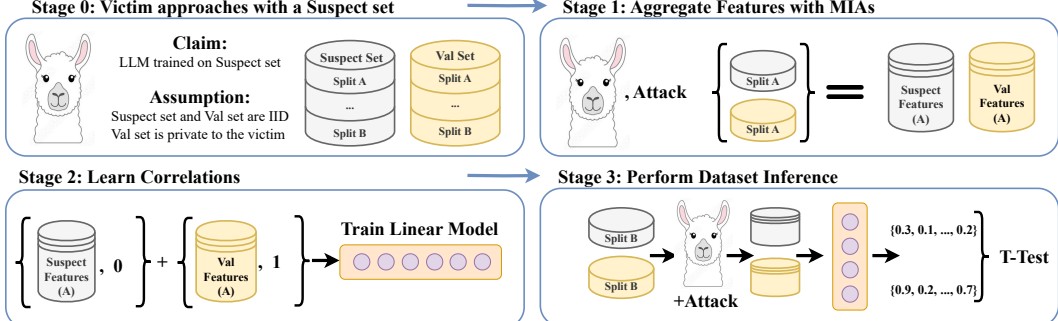

Figure 1: **LLM Dataset Inference.** *Stage 0: Victim approaches an LLM provider.* The victim's data consists of the suspect and validation (Val) sets. A victim claims that the suspect set of data points was potentially used to train the LLM. The validation set is private to the victim, such as unpublished data (e.g., drafts of articles, blog posts, or books) from the same distribution as the suspect set. Both sets are divided into non-overlapping splits (partitions) A and B. *Stage 1: Aggregate Features with MIAs.* The A splits from suspect and validation sets are passed through the LLM to obtain their features, which are scores generated from various MIAs for LLMs. *Stage 2: Learn Correlations (between features and their membership status).* We train a linear model using the extracted features to assign label 0 (denoting potential members of the LLM) to the suspect and label 1 (representing non-members) to the validation features. The goal is to identify useful MIAs. *Stage 3: Perform Dataset Inference.* We use the B splits of the suspect and validation sets, (i) perform MIAs on them for the suspect LLM to obtain features, (ii) then obtain an aggregated confidence score using the previously trained linear model, and (iii) apply a statistical T-Test on the obtained scores. For the suspect data points that are members, their confidence scores are significantly closer to 0 than for the non-members.

the training set increases, the success of membership inference degrades to random chance. *Is testing the membership of individual sentences for LLMs trained for a single epoch on trillions of tokens of text data feasible?* In our work, we first demonstrate that previous claims of successful membership inference for individual text sequences in LLMs [Mattern et al., 2023, Shi et al., 2024] are overly optimistic (Section 4). Our evaluation of the MIA methods for LLMs reveals a crucial confounder: they detect (temporal) distribution shifts rather than the membership of data points (as also concurrently observed by [Duan et al., 2024]). Specifically, we find that these MIAs infer whether an LLM was trained on a *concept* rather than an individual *sentence*. Even when the outputs of such MIAs (weakly) correlate with actual sentence membership, we find that they remain very brittle across sentences from different data distributions, and no single MIA succeeds across all. Based on our experiments, we conclude the discussion of MIAs with guidelines for future researchers to conduct robust experiments, highlighting the importance of using IID splits (between members and non-members), considering various data distributions, and evaluating false positives to mitigate confounding factors.

If membership inference attacks are so brittle, do content writers and private individuals have no recourse to claim that LLM providers unfairly trained on their data? As an alternative to membership inference, we advocate for a shift in focus towards dataset inference [Maini et al., 2021], which is a statistically grounded method to detect if a given *dataset* was in the training set of a model. We propose a new dataset inference method for LLMs that aims at detecting sets of text sequences by specific authors, thereby offering a more viable approach to dataset attribution than membership inference. Our method is presented in Figure 1. The motivation behind dataset inference stems from the observation that in the rapidly evolving discourse on copyright, individual data points have much less agency than sets of data points attributed to a particular creator; and the fact that more often than not, cases of unfair use emerge in scenarios when multiple such sequences or their clusters naturally occur. For instance, consider the Harry Potter series written by J.K. Rowling. Dataset inference tests whether a 'dataset' or a collection of paragraphs from her books was used for training a language model, rather than testing the membership of individual sentences alone. We also outline the specific framework required to operationalize dataset inference, including the necessary assumptions for the same.

We carry out our analysis of dataset inference using LLMs with known training and validation data. Specifically, we leverage the Pythia suite of models Biderman et al. [2023] trained on the Pile dataset Gao et al. [2020] (Section 5). This controlled experimental setup allows us to precisely analyze the model behavior on members and non-members when they occur IID (without any temporal shift) as

the training and validation splits of PILE are publicly accessible. Across all subsets, dataset inference achieves p-values less than 0.1 in distinguishing between training and validation splits. At the same time, our method shows no false positives, with our statistical test producing p-values larger than 0.5 in all cases when comparing two subsets of validation data. To its practical merit, dataset inference requires only 1000 text sequences to detect whether a given suspect dataset was used to train an LLM.

## 2   Background and Baselines

**Membership Inference.**   (MI) [Shokri et al., 2017]. The central question is: *Given a trained model and a particular data point, can we determine if the data point was in the model's training set?* Applications of MI methods span across detecting contamination in benchmark datasets [Oren et al., 2024, Shi et al., 2024], auditing privacy [Steinke et al., 2023], and identifying copyrighted texts within pre-training data [Shafran et al., 2021]. The field has been studied extensively in the realm of ML models trained via supervised learning on small datasets. The ability of membership inference in the context of large-scale language models (LLMs) remains an open problem. Recently, new methods [Mattern et al., 2023, Shi et al., 2024] have been proposed to close the gap and we present them in § 2.1.

**Dataset Inference.**   [Maini et al., 2021] provides a strong statistical claim that a given model is a derivative of its own private training data. The key intuition behind the original method proposed for supervised learning is that classifiers maximize the distance of *training examples* from the model's decision boundaries, while the *test examples* are closer to the decision boundaries since they have no impact on the model weights. Subsequently, dataset inference was extended from supervised learning to the self-supervised learning (SSL) models [Dziedzic et al., 2022] based on the observation that representations of the training data points induce a significantly different distribution than the representation of the test data points. We introduce dataset inference for large language models to detect datasets used for training.

### 2.1   Metrics for LLM Membership Inference

This section explores various metrics used to assess Membership Inference Attacks (MIAs) against LLMs. We study MIAs under gray-box access (which assumes access to the model loss, but not to parameters or gradients). The adversary aims to learn an *attack* function $A_{f_\theta} : \mathcal{X} \to \{0, 1\}$ that takes an input $x$ from distribution $\mathcal{X}$ and determines whether $x$ was in the training set $\mathcal{D}_{\text{train}}$ of the LM $f_\theta$ or not. Let us now describe the MIAs we use in our work.

**Thresholding Based.**   These MIAs leverage loss [Yeom et al., 2018] or perplexity [Carlini et al., 2021] as scores and then threshold them to classify samples as members or non-members. Specifically, the decision rule for membership is: $A_{f_\theta}(x) = \mathbb{1}[\mathcal{L}(f_\theta, x) < \gamma]$, where $\gamma$ is a selected pre-defined threshold. However, MIAs based solely on perplexity suffer from many false positives, where simple and predictable sequences that never occur in the training set can be labeled as *members*.

**MIN-K% PROB.**   As a remedy to the problem of predictability, Shi et al. [2024] proposed the MIN-K% PROB metric which evaluates the likelihood of the $K\%$ of tokens in $x$ that have the lowest probability, conditioned on the preceding tokens. Hence, this MIA ignores highly predictable tokens in the suspect sequence. The membership prediction is made by thresholding the average negative log-likelihood of these tokens. The input sentence $x$ is marked as included in pretraining data simply by thresholding the MIN-K% PROB result: $A_{f_\theta}(x) = \mathbb{1}[\text{MIN-K\% PROB}(x) < \gamma]$.

**Perturbation Based.**   The central hypothesis behind Perturbation-based MIAs is that a sample that an LLM saw during training should have a lower perplexity on its original version $(x)$, as opposed to a perturbed version of the same $(\tilde{x})$. Formally, the membership attack is defined as $A_{f_\theta}(x) = \mathbb{1}\left[\mathcal{P}f_\theta(x)/\mathcal{P}f_\theta(\tilde{x}) < \gamma\right]$, for a threshold $\gamma$. In our work, we investigate various forms of perturbations such as (1) white-space perturbation, (2) synonym substitution [Mattern et al., 2023], (3) character-level typos, (4) random deletion, and (5) changing character case.

**DetectGPT.**   This is a special case of perturbation-based MIAs, originally used to detect machine-generated text [Mitchell et al., 2023]. The key difference is that perturbations to the input are made using an external language model that infills randomly masked-out spans of the original input. It then compares the log-probability of $x$ with expected value of the same from multiple infilled neighbors $\tilde{x}_i$.

**Reference Model Based.** These methods compare the perplexity ratio between a suspect model and a reference model on a given string. The suspect model may have seen the string during training, while the reference model has not. The corresponding MIA is: $A_{f_\theta}(x) = \mathbb{1}[\mathcal{L}(f_\theta, x) < \mathcal{L}(f'_\theta, x)]$, where $f'_\theta$ is the reference model. In our work, we use the SILO [Min et al., 2023], Tinystories-33M [Eldan and Li, 2023], Tinystories-1M [Eldan and Li, 2023], and Phi-1.5 [Li et al., 2023] models as reference models. Notably, these models were not trained on general web data. In particular, the Phi-1.5 and Tinystories models were trained on synthetic data generated by GPT models, and the SILO model was trained on data that is freely licensed for training.

**zlib Ratio.** Another simple MI baseline uses the *zlib library* [Gailly and Adler, 2004], where a potential member has a low ratio of the model's perplexity to the entropy of the text, which is computed as the number of bits for the sequence when compressed with the zlib library: $A_{f_\theta}(x) = \mathbb{1}[\mathcal{P}_{f_\theta}(x)/zlib(x) < \gamma]$ [Carlini et al., 2021]. The idea is that a model trained on a dataset will have low perplexity for its members because it was optimized for them, unlike the zLib algorithm, which was not tailored to the training data.

## 3 Problem Setup

LLMs train on trillions of tokens, and the sizes of the training sets are only likely to increase [Met, dbr]. To increase training efficiency (in terms of time, financial costs, and environmental impact), improve performance, and decrease the risk of privacy leakage, many LLM practitioners deduplicate their pre-training data [Biderman et al., 2023, Carlini et al., 2021, Lee et al., 2022]. In our work, we ask this question: *How to detect if a given dataset was used to train an LLM?* and propose the idea of dataset inference for LLMs.

**Access Levels.** In the *black-box* setting, we assume an input-output access to an LLM along with access to model loss, hence we are not allowed to inspect individual weights or hidden states (*e.g.,* attention layer parameters) of the language model. This threat model is realistic in the case of LLM's users since many language models can be accessed through APIs that provide limited visibility into their inner workings. For instance, OpenAI [Ope] offers API access to GPT-3 and GPT-4, while Google [Gem] offers Gemini, without revealing the full architecture of the models or training methodology. The *gray-box* access, commonly assumed for MIAs, additionally assumes that we can obtain the perplexity or the loss values from an LLM, however, no additional information such as model weights or gradients. In the *white-box* access, we assume full access to the model, where we can inspect model weights.

**Operationalizing Dataset Inference.** Dataset inference for LLMs serves as a detection method for data used to train an LLM. We consider the following three actors during a dispute:

1. *Victim (V).* We consider a victim creator whose private or copyrighted content was used to train an LLM without explicit consent. The actor is presumed to have only black-box access to the suspect model, which limits their ability to evaluate if their dataset was used in the LLM's training process.
2. *Suspect (A).* The suspect (or potential adversary in this case) is an LLM provider who may have potentially trained their model on the victim's proprietary, or private data.
3. *Arbiter.* We assume the presence of an arbiter, *i.e.,* a third-trusted party, such as law enforcement, that executes the dataset inference procedure. The arbiter can obtain gray-box access to the suspect LLM. For instance, in scenarios when API providers only give black-box access to users, legal arbiters may have access to model loss to perform MIAs.

**Scenario.** Consider a scenario where a book writer discovers that their publicly available but copyrighted manuscripts have been used without their consent to train an LLM. The writer, the *victim* $V$ in this case, gathers a small set of text sequences (say 100) from their manuscripts that they believe the model was trained on. The *suspect* $A$ in this scenario is the LLM provider, who may have included the writer's published work in their training data without obtaining explicit permission. The provider is under suspicion of potentially infringing on the writer's manuscripts. An *arbiter*, such as a law enforcement agency, steps in to resolve the dispute. The arbiter obtains gray-box access to the suspect LLM, allowing them to execute our dataset inference procedure and resolve the dispute. By performing dataset inference (as depicted in Figure 1), the arbiter determines whether the writer's published manuscripts were used in the training of the LLM. This process highlights the practical application and significance of dataset inference in safeguarding the rights of artists.

**Notation.** We consider $x$ to be an input sentence with $N$ tokens $x = x_1, x_2, ..., x_N$ and $f_\theta$ is a Language Model (LM) with parameters $\theta$. We can compute the probability of an arbitrary sequence $f_\theta(x_1, ..., x_n)$, and obtain next-token $x_{n+1}$ predictions. For simplicity, assume that the next token is sampled under greedy decoding, as the next token with the highest probability given the first $n$ tokens.

## 4 Failure of Membership Inference

We demonstrate that the challenge of successfully performing membership inference for large language models (LLMs) remains unresolved. This problem is inherently difficult because LLMs are typically trained for a *single epoch* on trillions of tokens of web data. In their work, Maini et al. [2021] demonstrated a near impossibility result (Theorem 2), suggesting that as the size of the training set increases, the success rate of any MIA approaches 0.5 (as good as a coin flip). While this was shown in a simplified theoretical model, we assess how this holds up for contemporary LLMs with billions of parameters. As a demonstrative example, we consider the most recent (and supposedly best performing) work that proposed the MIN-K% PROB [Shi et al., 2024] membership inference attack, alongside a dedicated dataset to facilitate future evaluations. In their work, they show that this method performs notably better than other MIAs such as perplexity thresholding and DetectGPT that they benchmark their work against.

**Temporal Shift and the Need for IID Analysis.** The evaluation dataset used to showcase the success of MIN-K% PROB was the WikiMIA dataset, a dataset constructed using spans of Wikipedia articles written before (train set) and after the cut-off year 2023 (validation set). This was chosen considering the training of the Pythia models [Biderman et al., 2023], which was based on scrapes of Wikipedia before 2023. Note that such an evaluation setup naturally has the potential confounder of a temporal shift in the concepts in data before and after 2023. Any article written after 2023 was naturally a non-member of the Pythia models, and those written before 2023 were considered members. However, with changing times, we also encounter temporal shifts in writing styles and concepts in the Wikipedia dataset. This raises concerns if membership tests using WikiMIA actually assess membership of a particular data point, or of that concept/style. A similar question was concurrently asked by Duan et al. [2024], who independently showed that MIAs are only successful because of the temporal shift in such datasets.

To critically assess the robustness of the MIN-K% PROB method, we conducted an exploration using the Pythia models and their (original) train and validation splits that come from the PILE [Gao et al., 2020] dataset, as provided by the authors during pre-training. This facilitates a confounder-free evaluation of the capabilities of membership inference attacks. In particular, the PILE dataset has more than 20 different domain-specific subsets with their own training and validation splits, such as Arxiv, Wikipedia, OpenWebtext to name a few. Some of the key observations from our experiments on the PILE were (Figure 2a):

1. Contrary to performance on the WikiMIA dataset where MIN-K% PROB metric achieved an AUC close to 0.7, the method got an AUC close to 0.5 when tested on IID train and validation splits of Wikipedia from the PILE dataset, hinting at a performance akin to random guessing.
2. We found that the method shows very high variance in AUC between different random subsets of the training and validation sets of the PILE dataset, oscillating between 0.4 and 0.7.
3. Results on Arxiv and OpenWebText2 subsets of the PILE show AUC values near 0.4, suggesting that MIN-K% PROB suffers from false positives, labeling validation set examples as members.

**False Positive Assessment by Reversing Train and Val Sets.** *Do membership inference attacks actually test membership?* To answer this question, we do the following modification to the WikiMIA setup: For every sentence in the pre-2023 subset of Wikipedia, we replace it with a sentence from the validation set for Wikipedia as given in the PILE dataset. We keep the post-2023 Wikipedia subset as it is. On the one hand, since Pythia models were trained before 2023, it is clear that they never trained on data on Wikipedia from pages written after 2023. On the other hand, we also know that the validation set of the PILE dataset was not trained on and was also deduplicated from the train set. We now perform the same membership inference test on these two data splits Wikipedia Val (as the now designated 'suspect' set) versus Wikipedia post-2023 (as the supposed unseen set). Remarkably, the method demonstrates an extremely high AUC of 0.7 in labeling examples from the suspect (validation set) as members of the training set (Figure 2b). This confirms that these membership inference attacks (such as MIN-K% PROB) only distinguish between concepts across different temporal phases rather than verifying specific data membership, which they were originally designed for.

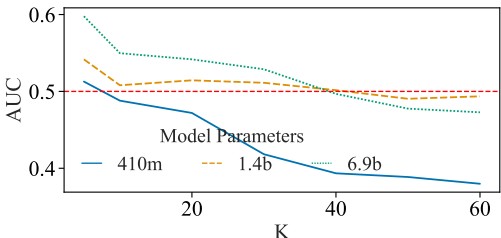
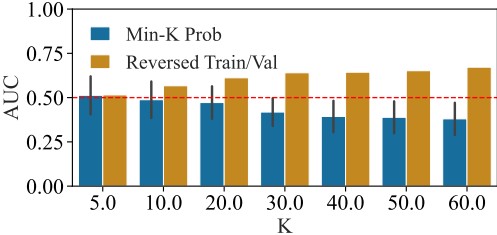

(a) Performance for different model sizes.

(b) False Positives when reversing train/val sets.

Figure 2: **Comparative analysis of the MIN-K% PROB [Shi et al., 2024].** We measure the performance (a) across different model sizes and (b) the observed reversal effect. The method performs close to a random guess on non-members from the Pile validation sets.

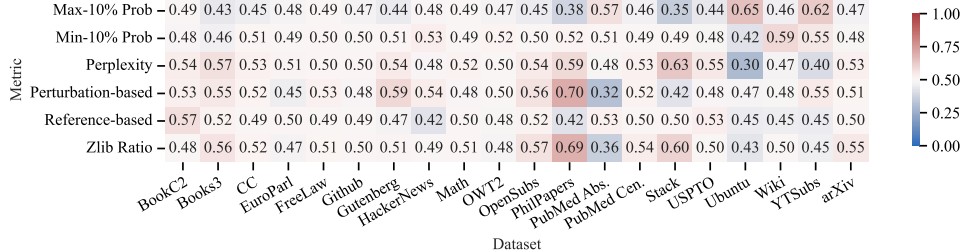

Figure 3: **Performance of various MIAs on different subsets of the Pile dataset.** We report 6 different MIAs based on the best performing ones across various categories like reference based, and perturbation based methods (Section 2.1). An effective MIA must have an AUC much greater than 0.5. Few methods meet this criterion for specific datasets, but the success is not consistent across datasets.

**No single MIA works across distributions.** Now, we further expand our experimentation across multiple different membership inference attacks outlined in Section 2.1, across 20 different subsets of the PILE dataset. The goal is to analyze if there is any MIA that consistently performs well across all such distributions. In Figure 3, we show a heatmap of the performance of various (selected) MIA methods across different distributions of the PILE (refer to Appendix C for full results). While some MIAs perform well and achieve high AUC (such as synonym substitution on PhilPapers), the same methods have an AUC of less than 0.5 on the next dataset of Pubmed Abstracts. These results consolidate the finding that no single MIA for LLMs works across all datasets, and we need to potentially find methods that adapt the choice of metric to the distribution. In Section 5, we will leverage a (selective) combination of different MIAs to improve over the performance of any single MIA in order to perform successful LLM Dataset Inference.

**Guidelines for Future Research.** Based on our observations in this section, we outline four important practices for future research in membership inference to enable sound experimentation and inferences. In particular, (1) assessment for membership inference must be done in an IID setup where train and validation splits are from the same distribution, (2) experiments must be repeated over multiple random splits of the datasets, (3) experiments must be performed over multiple data distributions (4) careful experimentation must be done on both false positives and false negatives to ensure MIAs do not wrongly label non-members as members.

## 5 LLM Dataset Inference

Dataset inference builds on the idea of membership inference by leveraging distributional properties to determine if a model was trained on a particular dataset. While MIAs operate at the instance level—aiming to identify whether each example was part of the training data. In the previous sections, we have shown that MIAs often yield signals that is close to random in determining example membership. However, if we achieve even slightly better than random accuracy in inferring membership, we can aggregate these attacks across multiple examples to perform a statistical test. This test can then distinguish between the distributions of the model's training and validation sets. In the context of LLM dataset inference, we combine all the MIA methods discussed in Section 2.1.

## 5.1 Procedure for the LLM Dataset Inference

We describe the procedure for LLM dataset inference in four stages (see also visualization in Figure 1). Recall the initial example of a book writer who suspects that a portion of their books was trained on. We use this as a running example to describe the four stages of LLM dataset inference.

**Stage 0: Victim approaches an LLM provider.** A victim (author) $\mathcal{V}$ approaches an arbiter with a claim of ownership over data (book) that they suspect a model trainer or adversary $\mathcal{S}$ utilized. This stage involves the arbiter validating if the claim by $\mathcal{V}$ satisfies the assumptions under which dataset inference operates, that is, they provide an IID set of data that they suspect was trained on, and an equivalent dataset that $\mathcal{S}$ could not have seen, denoted as the validation set. This can, for instance, happen when authors have multiple drafts of a book chapter, and only one of the drafts makes it to the actual print. In such a case, $\mathcal{V}$ claims that $\mathcal{S}$ trained on the published version of their book, because $f_\theta(\mathcal{S})$ responds *differently* to the final versus rejected drafts of the book. Both suspect and validation sets are divided into non-overlapping splits (partitions) A and B. We will use $A_{\text{val}}$, $A_{\text{sus}}$ in Stages 1, 2 and $B_{\text{val}}$, $B_{\text{sus}}$ in Stage 3 to actually perform the ownership assessment.

**Stage 1: Aggregate Features with MIAs.** This step involves aggregating scores from various MIA methods described in the previous Section 2.1. $A_{\text{val}}$, $A_{\text{sus}}$ sets are passed through the LLM under question to obtain their features derived from MIAs. Note that we use an aggregation of all the MIA methods discussed in the previous sections to create a single feature vector. In the next step, we will determine which MIAs are useful for identifying dataset membership for the given distribution.

**Stage 2: Learn MIA correlations.** In this stage, we train a linear regressor to learn the importance of weights for different MIA attacks to use for the final dataset inference procedure. Across each MIA feature value, we first modify the top 5% outliers by changing their values to the mean of the distribution. This step is crucial to prevent issues in Step 3, where the model might learn skewed correlations due to a few outlier samples. We then pass the data through a linear regression model to learn weights for each feature. All 'suspect' samples in $A_{\text{sus}}$ are labeled as 0, and all validation samples in $A_{\text{val}}$ are labeled as 1. A regressor is trained to predict the label given the samples, effectively learning the correlation between the features and their membership status.

**Stage 3: Perform Dataset Inference.** We use $B$ splits of the suspect and validation sets, holding out up to 1000 samples in these splits for ownership assessment. Each sample is assigned a membership value using a trained linear classifier. These values are used to perform a statistical t-test to determine if the suspect set was used in training the model. We then report whether the model was trained on the suspect dataset based on the t-test results. For members of the suspect set, their confidence scores are significantly closer to 0 compared to non-members. The null hypothesis ($H_0$) is that the suspect dataset was not used for training. Assume that $\mu_{\mathcal{M}(B_{\text{sus}})}$ and $\mu_{\mathcal{M}(B_{\text{val}})}$ are the mean membership values of the suspect and validation sets, respectively. Then, $H_0$ and $H_1$ (alternate hypothesis) are:

$$H_0 : \mu_{\mathcal{M}(B_{\text{sus}})} \geq \mu_{\mathcal{M}(B_{\text{val}})}; \qquad H_1 : \mu_{\mathcal{M}(B_{\text{sus}})} \leq \mu_{\mathcal{M}(B_{\text{val}})}. \tag{1}$$

**Combining p-values for Dependent Tests.** To assess the significance of the results, we performed multiple t-tests using 10 different random seeds to obtain various splits of examples between A and B sets. Since the subsets had overlapping examples, the statistical tests are dependent [Vovk and Wang, 2020], and p-values must be aggregated accordingly [Brown, 1975, Kost and McDermott, 2002, Meng, 1994, Rüschendorf, 1982]. Based on the observation in Blakesley et al. [2009], the Sidak-method [Šidák, 1967] for combing p-values is typically conservative and helps avoid Type-1 errors. Let $p_1, p_2, \ldots, p_n$ denote the p-values obtained from the $n$ t-tests, then the aggregated $p$-value:

$$p_{\text{combined}} = 1 - \exp\left(\sum_{i=1}^{n} \log(1 - p_i)\right) \tag{2}$$

**Score Aggregation.** To aggregate scores from different MIAs, we **(i) normalize feature values** to ensure that all features aggregated across various membership inference attacks are on a comparable scale. Then, we **(ii) adjust values of outliers** before learning correlations with the classifier by replacing the top and bottom 2.5% of outlier values with the mean of that (normalized) feature. Finally, we **(iii) remove outliers before performing t-test** in Stage 3 once we have a single membership value from the regressor outputs for each sample in the $B$ splits of the suspect and validation sets. Once again we remove the top and bottom 2.5% of outlier.

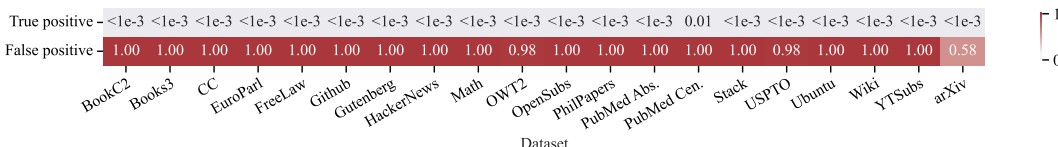

Figure 4: **p-values of dataset inference** By applying dataset inference to Pythia-12b models with 1000 data points, we observe that we can correctly distinguish train and validation splits of the PILE with very low p-values (always below 0.1). Also, when considering false positives for comparing two validation subsets, we observe a p-value higher than 0.1 in all cases, indicating no false positives.

## 5.2 Assumptions for Dataset Inference

In order to operationalize dataset inference, we must obey certain assumptions on both the datasets (points 1 and 2 below), and the suspect language model (point 3 below).

1. The suspect train set and the unseen validation sets should be IID. This prevents the results from being confounded due to distribution shifts (such as temporal shifts in the case of WikiMIA).
2. We must ensure no leakage between the (suspected) train and (unseen) validation sets. The validation set should be strictly private, and only accessible to the victim.
3. We need access to the output loss of the suspect LLM in order to perform various MIAs.

## 5.3 Experimental Details

**Datasets and Architectures.** We perform dataset inference experiments on all 20 subsets of the PILE. For experiments with false positives, we split the validation sets into two subsets of 500 examples each. In all other experiments, we compare 1000 examples of train and validation sets of the PILE [Gao et al., 2020]. We perform dataset inference on models from the Pythia [Biderman et al., 2023] family at 410M, 1.4B, 6.9B, and 12B scales. These open-source models allow us to know exactly which examples trained on.

**MIAs used.** In our experiments, we aggregate 52 different Membership Inference Attacks (MIAs) in Stage 1 (many of which are overlapping and only differ in whether they capture the perplexity or the log-likelihood, or contrast the ratios or differences of model predictions). For the linear regression model trained in Stage 2, we train for 1000 updates over the data using simple weights over the 52 features. A total of 1000 examples are saved for training the regressor to learn correlations for stage 2, except in the false positive experiments where we use half the data. A complete list of all the MIAs used in our work is present in Appendix C.

## 5.4 Analysis and Results with Dataset Inference

We analyze the performance of LLM dataset inference on the Pythia suite of models [Biderman et al., 2023] trained on the Pile dataset [Gao et al., 2020]. We separately perform dataset inference on each and every subset of the PILE using the provided train and validation sets, and report the p-values for correctly identifying the training dataset. Before diving into various design choices, the key result is that dataset inference reliably finds the training distribution in all subsets of the PILE. (Figure 4). For the analysis of false positives, we carry out dataset inference on two splits of the validation set for each subset of the PILE. Since neither of the validation subsets was used to train the model $f_\theta$, the returned p-values should be (and are) significantly above the selected threshold of 0.1 for any useful attribution framework. It is worth noting that the p-values for these tests are often remarkably low in the order of $10^{-30}$ and lower, suggesting high confidence in attributing dataset ownership. When contrasted with the lack of reliability of membership inference, dataset inference indeed shows great promise for future discourse on inspecting training datasets. We will now dive deeper into various ablations and results around dataset inference such as the features (membership inference attacks) chosen by the regressor, choice of pre-processing function, change in performance of dataset inference with model size, data duplication and number of permitted queries.

**Feature Selection.** For each domain, we find that the most important metrics are different. Hence, the linear classifier is essential to appropriately determine the importance of each feature for determining per example membership, based on the dataset statistics. We present the results for a subset of MIAs in Figure 5a (all MIAs in Appendix Figure 8). For example, while the Perturbation-based metric is necessary to be present for the CC dataset, it is not useful for the OWT2 dataset, which instead

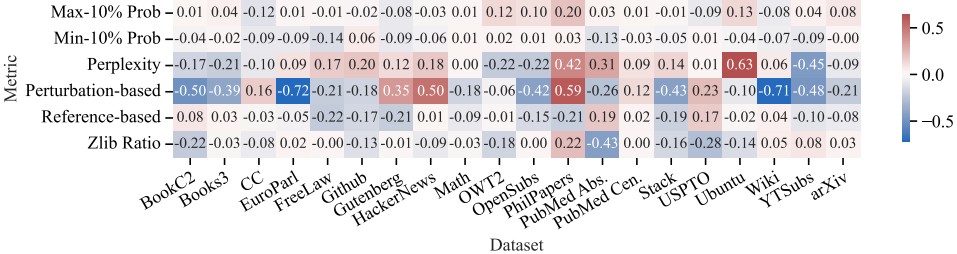

(a) **Feature Selection.** A positive value indicates a correlation of the feature with the validation set, while a negative one denotes a correlation with the train set.

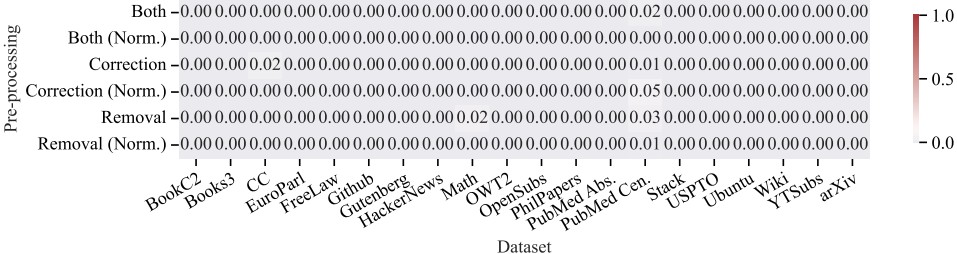

(b) **Feature pre-processing.** We present the p-values for a given dataset using differently pre-processed features.

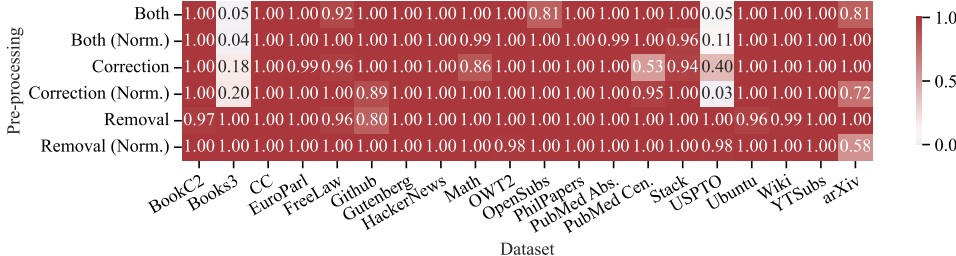

(c) **Analysis of false-positives.** A p-value smaller than 0.1 indicates a false positive.

Figure 5: **Ablation study for dataset inference.** We analyze which features based on or derived from the previous membership inference methods increase the success of dataset inference. (a) Our results indicate that no single feature contributes consistently, thus we need a linear model to selectively aggregate their impact on the final outputs from dataset inference. (b) Given the selected features, we consider different ways of how to pre-process them before building the classifier. The proposed method (denoted as Removal (Norm.)) removes outliers and normalizes the feature values. (c) We evaluate the selected and pre-processed features using suspect set that come from the validation data. We do not observe any false positives as shown in the last row in (c).

requires the Perplexity metric. Dataset inference automatically learns which MIAs are positively correlated with a given distribution. The linear regressor can be trained quickly on a CPU since it is only learning a weight assignment for each feature. Now, we investigate which MIAs get selected by dataset inference by analyzing the importance weights given by our linear regression to various MIAs.

**Feature Pre-Processing.**    Considering the chosen features, we explore various pre-processing techniques to apply before training the linear regressor. The selected approach, referred to as Removal (Norm.) in Figure 5b, involves eliminating outliers and normalizing the feature values. We tried other approaches such as mean correction, and outlier clipping, but we found that these approaches make the dataset inference procedure less reliable by artificially modifying the score distributions and modifying the feature correlations learned by the linear model. Its effect can be seen through the occurrence of false positives for some of the datasets.

**Number of Queries.**    We analyze the number of queries that have to be executed against the tested model $f_\theta$ to determine if a given dataset was used for training. We present the results in Figure 6a. It can be seen more than half of datasets only require about 100 points, while 1000 points are sufficient to obtain p-values smaller than the significance threshold of 0.1 for all datasets.

**Size of LLMs and Training Set Deduplication.**    By studying the Pythia suite of models [Biderman et al., 2023] which are trained on the same dataset, we observe the success of dataset inference is

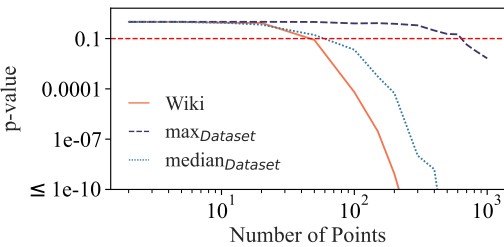 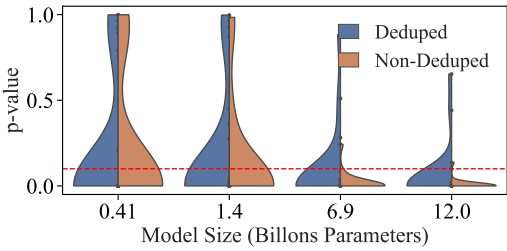

(a) Dataset inference success v.s. number of data points.    (b) Dataset inference success v.s. LLM size.

Figure 6: **Ablation studies for the amount of data and model size.** In (a), we plot the maximum and median p-values across all datasets, alongside the p-value of Wikipedia, as a function of the number of data points. In (b), a violin plot is made to show the distribution of p-values of the datasets with respect to the number of model parameters. Observe that dataset inference is more successful with more data and larger LLMs. It is also noteworthy that (a) dataset inference for a majority of datasets is accurate with less than 100 points, and (b) it is more accurate with respect to the non-deduplicated models that are trained on datasets with duplicated points. We hypothesize this is because the membership signal for most MIAs becomes stronger with the duplication of data. Deduped denotes a version of the Pile dataset where the documents are deduplicated within, and across the data splits (train/test/validation). Non-Deduped is the original version of Pile without any deduplication.

positively correlated with the number of parameters in the LLMs. We present this result in Figure 6b as a violin plot to allow for visualizing distributions of the datasets' p-values. It can be seen as the size of the model increases, the p-value distribution concentrates below the threshold of $0.1$. This correlation can be explained by the phenomenon that memorization by LLMs increases as their parameter size increases [Carlini et al., 2021], which provides a stronger signal for the intermediate MIAs responsible for dataset inference to succeed. We also contrast the models trained on deduplicated or non-deduplicated training sets when we are only allowed 500 query points. Observe that while the aforementioned trend holds for both kinds of models, the p-value distribution is more concentrated below 0.1 for the non-deduplicated models. Following the same explanation as above, this also indicates that memorization is more severe when some training data is duplicated, allowing various membership inference attacks to have a stronger signal.

## 6    Discussions

**Membership Inference for LLMs.**   In this work, we question the central foundations of research on membership inference in the context of LLMs trained on trillions of tokens of web data. Our findings indicate that current membership inference attacks for LLMs are as good as random guessing. We demonstrate that past successes in MIAs are often due to specific experimental confounders rather than inherent vulnerabilities. We provide guidelines for future researchers to conduct robust experiments, emphasizing the use of IID splits, considering various data distributions, assessing false positives, and using multiple random seeds to avoid confounders.

**Shift to LLM Dataset Inference.**   Historically, membership inference focused on whether an individual data point was part of a training dataset. Instead, we aggregate multiple data points from individual entities, forming what we now consider a dataset. In our work, we have not only put thought towards the scientific framework of dataset inference but also the ways it will operationalize in real-world settings, for instance, through our running example of a writer who suspects that their books were trained on. Our research demonstrates that LLM dataset inference is effective in minimizing false positives and detecting even minute differences between training and test splits of IID samples.

**Limitations.**   A central limitation to dataset inference is the assumptions under which it can be performed. More specifically, we require that the training and validation sets must be IID, and the validation set must be completely private to the victim. While this may appear elusive a priori, we outline concrete scenarios to show how these sets naturally occur. For instance, through multiple drafts of a book, until one gets finalized. The same applies to many artistic and creative uses of LLMs across language and vision today. In terms of data and model access, we assume that the victim or a trusted third party, such as law enforcement, is responsible for running the dataset inference so that there are no privacy-related concerns. This will require the necessary legal framework to be brought in place, or otherwise suspect adversaries may deny querying their model altogether.

# 7 Acknowledgements

We would like to acknowledge our sponsors, who support our research with financial and in-kind contributions: Amazon, Apple, CIFAR through the Canada CIFAR AI Chair, Meta, NSERC through the Discovery Grant and an Alliance Grant with ServiceNow and DRDC, the Ontario Early Researcher Award, the Schmidt Sciences foundation through the AI2050 Early Career Fellow program, and the Sloan Foundation. Resources used in preparing this research were provided, in part, by the Province of Ontario, the Government of Canada through CIFAR, and companies sponsoring the Vector Institute. Pratyush Maini was supported by DARPA GARD Contract HR00112020006. Adam Dziedzic was supported by the German Federal Ministry of Education and Research (BMBF) under funding number 16KIS2114K and by the German Research Foundation (DFG) within the framework of the Weave Programme supporting the project on "Protecting Creativity: On the way to Safe Generative Models" under project number 545047250. Responsibility for the content of this publication lies with the authors.

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

# A  Broader Impact

The use of large amounts of text scrapes from the web to train large language models (LLMs) has recently sparked significant privacy concerns. Including personally identifiable information (PII) and copyrighted material in these datasets has resulted in legal disputes, such as the lawsuit between The New York Times and OpenAI, among others. These disputes emphasize the problem of using copyrighted content without proper attribution, potentially infringing on creators' rights and discouraging future artistic work due to the lack of financial compensation for publicly available content. This scenario underscores the technical challenge of identifying data used to train machine learning models. Despite the legal uncertainties, this task is essential for understanding LLM behavior and ensuring data privacy. In our work, we propose a new *dataset inference* method to accurately identify the datasets used to train large language models. Our LLM dataset inference method is statistically grounded and we thoroughly evaluate our approach.

# B  Compute

To aggregate various membership inference attacks, we required performing forward passes for both suspect and reference models. We need enough GPU memory to load a model. To allow for large batch forward passes, we utilized NVIDIA A6000 48GB machines for aggregating such metrics. We used a total of 4 machines at any given point to speed up the aggregation of metrics.

# C  Additional Experiments

**Setup.** Note that for the perturbation-based MIAs (*e.g.,* [Mattern et al., 2023]), we use the implementation of the perturbations from the NLAugmenter library Dhole et al. [2023].

**Performance of MIAs.** We provide an extended version of the performance of various MIAs on different subsets of the Pile dataset in Figure 7. For an MIA to be effective, it must achieve an AUC significantly higher than 0.5. Only a few methods meet this standard for certain datasets, and their success is not consistent across different datasets. We note that, for example, the DetectGPT [Bao et al., 2024] performs significantly better on a few datasets, for example, bookcorpus2 and books3, than any other metric.

**Feature Selection.** We provide an extended version of the importance of features used in dataset inference in Figure 8.

**Additional ablation studies on model size, processing technique, and number of data points.** We analyze the number of required data points for successful dataset inference, with processing techniques used in dataset inference and different sizes of LLMs, in Figure 9 and Figure 10 respectively. To visualize the distribution of p-values impacted by the number of data points and size of the models, violin plots are presented in Figure 11.

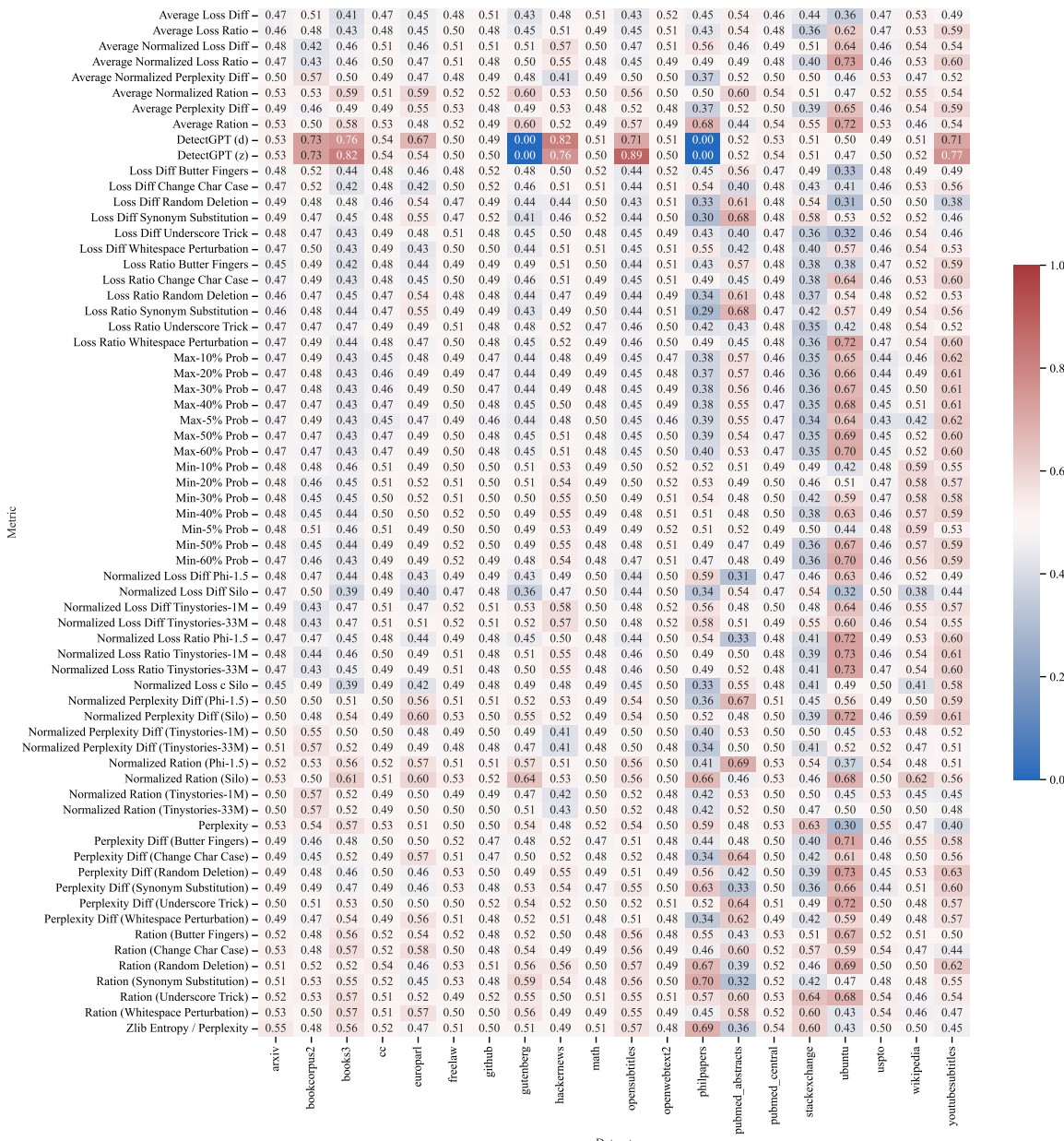

Figure 7: Extended version of Figure 3 with more types of membership inference attacks. The results are consistent: none of the membership inference attacks can consistently achieve high ROAUC across different datasets.

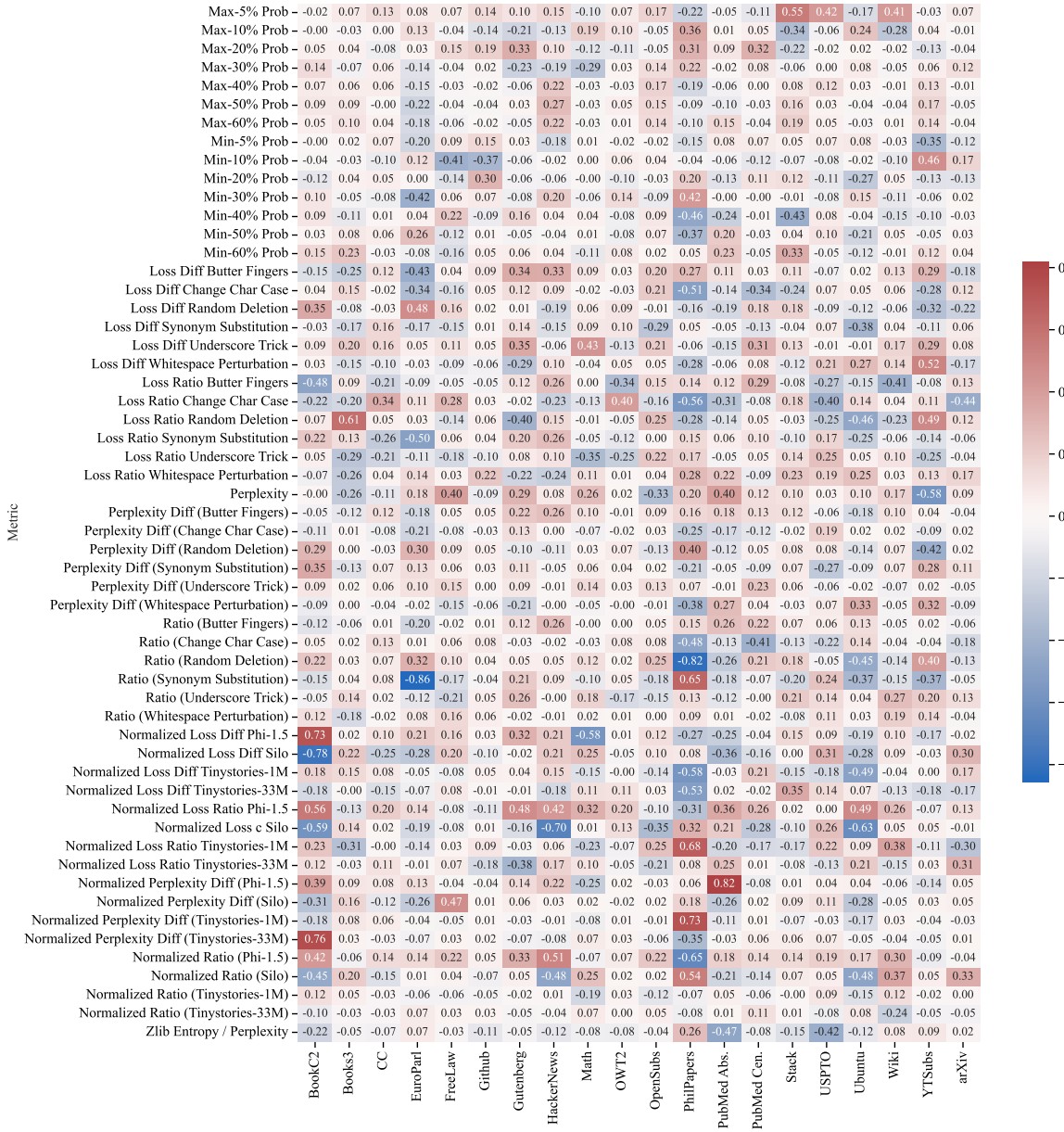

Figure 8: Extended version of Figure 5a with more features that we leverage for the LLM dataset inference. The results are consistent—one can see most features contribute positively to dataset inference for some datasets, while negatively to other datasets.

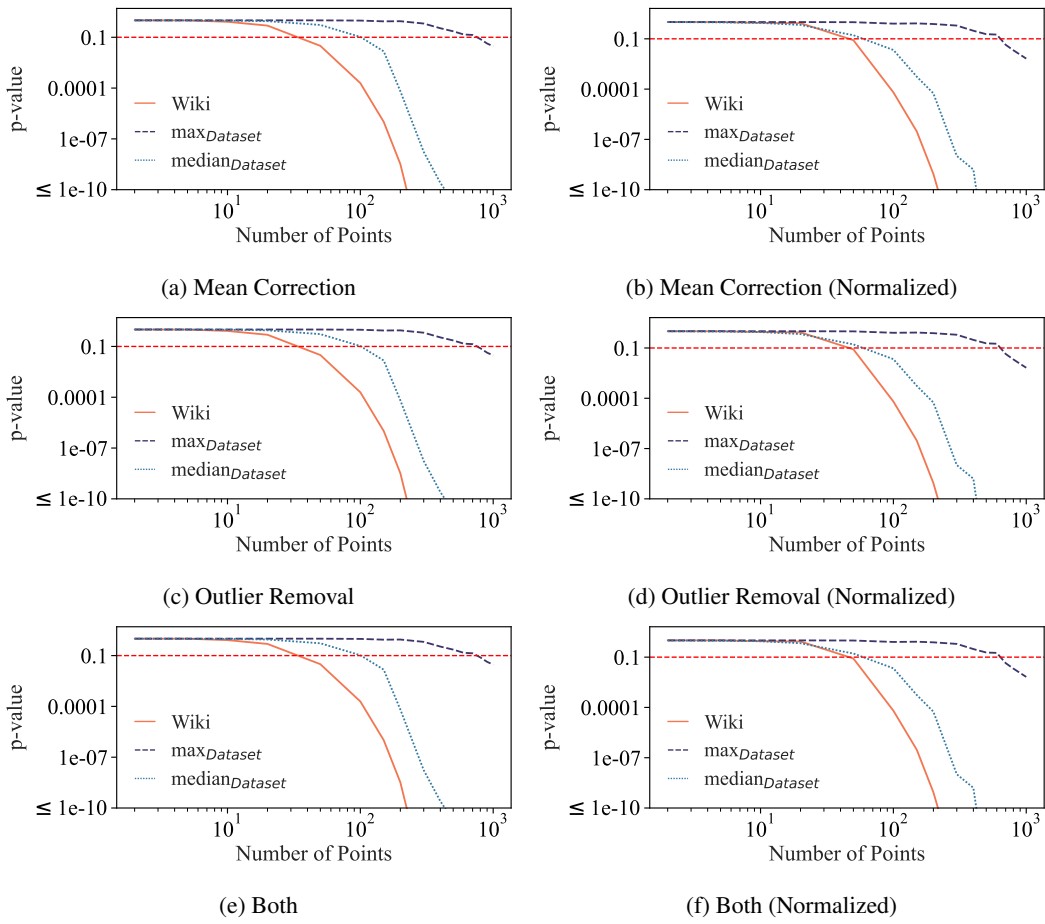

(a) Mean Correction

(b) Mean Correction (Normalized)

(c) Outlier Removal

(d) Outlier Removal (Normalized)

(e) Both

(f) Both (Normalized)

Figure 9: Extended version of Figure 6a with different pre-processing techniques used in dataset inference. Similarly, the three curves correspond to the median, and maximal p-value across the datasets in Pile, and the p-value of Wikipedia dataset. It is noteworthy that the processing techniques (see details in Section 5.3) do not impact the median and Wikipedia curves significantly. However, normalization has a positive contribution to the max p-value across different datasets.

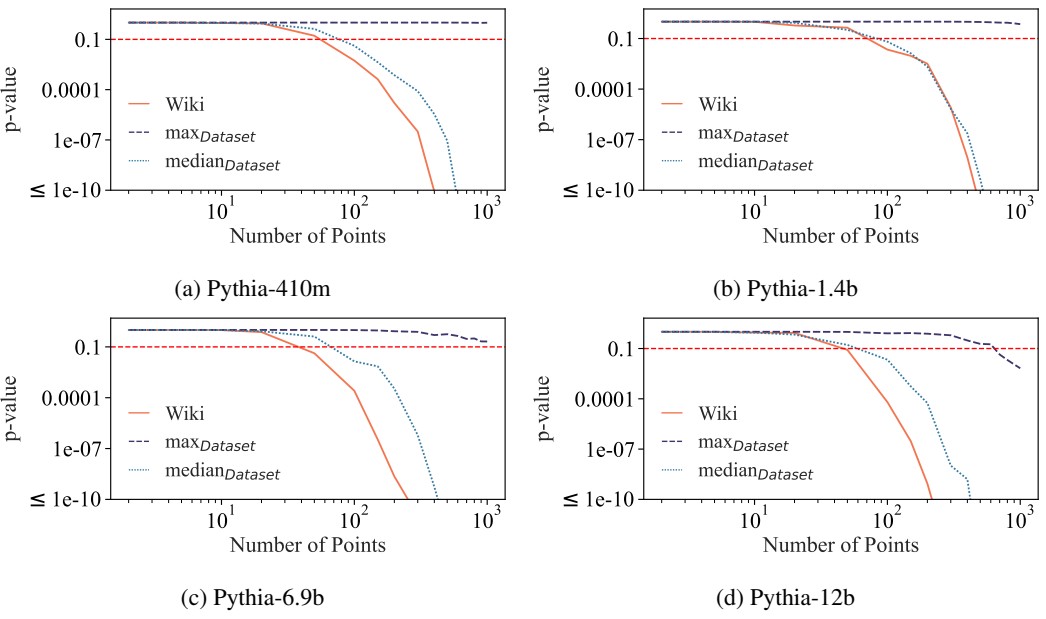

(a) Pythia-410m

(b) Pythia-1.4b

(c) Pythia-6.9b

(d) Pythia-12b

Figure 10: Extended version of Figure 6a with different model sizes. Similarly, the three curves correspond to the median, and maximal p-value across the datasets in Pile, and the p-value of Wikipedia dataset. An interesting observation is that the curves for median p-values and Wikipedia's p-value stay similar with respect to models of different sizes. However, this is not the case for the max p-value curve. This indicates that dataset inference does not rely on a large number of data points or model parameters for most datasets, whereas they may be necessary for some particular datasets.

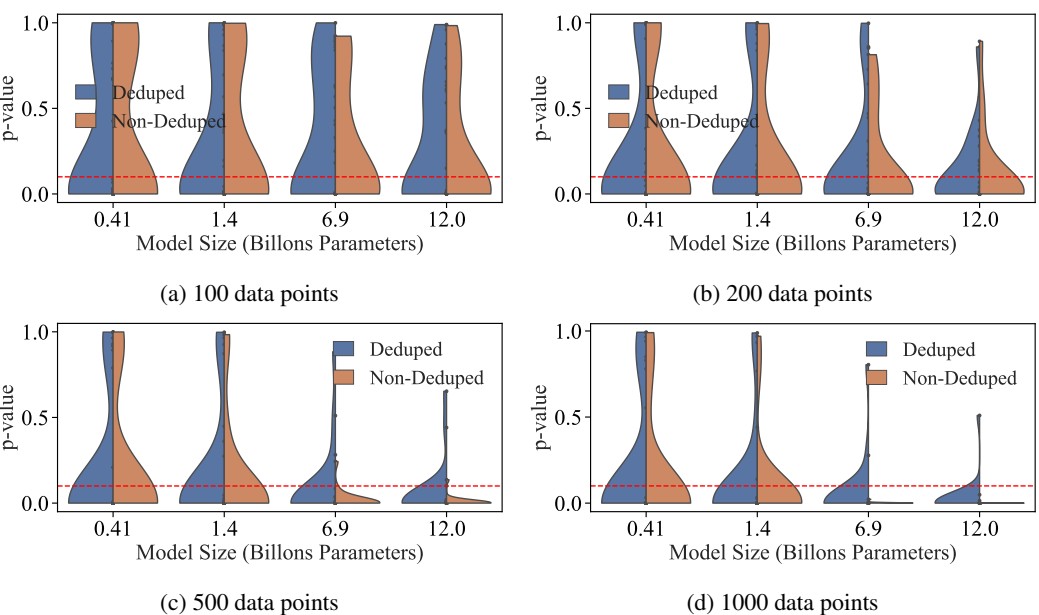

(a) 100 data points

(b) 200 data points

(c) 500 data points

(d) 1000 data points

Figure 11: Extended version of Figure 6b with different numbers of data points used in dataset inference. As expected, a larger number of data points allow the p-values to be more concentrated below the threshold, especially when the model size is large.

