# OpenReview forum: "LLM Dataset Inference: Did you train on my dataset?"
_NeurIPS.cc/2024/Conference — NeurIPS 2024 poster_

### Official Review · Reviewer_G1MR · 2024-06-15

**Soundness:** 3
**Presentation:** 3
**Contribution:** 3
**Rating:** 5
**Confidence:** 4

**Summary:**

The paper addresses the limitations of traditional membership inference attacks in identifying if specific text sequences belong to the training data of LLMs. The authors highlight the inadequacies of MIAs and propose a novel dataset inference method. This method focuses on detecting entire datasets used in model training, rather than individual strings, combining multiple MIA metrics to accurately distinguish between training and test datasets with statistically significant results and no false positives. This approach promises to enhance the legality and ethicality of training LLMs.

**Strengths:**

1. The paper provides a compelling argument for shifting the focus from individual string-based MIAs to dataset-based inference methods.
2. The paper is very well written

**Weaknesses:**

1. The author demonstrates the failures of MIA methods by assessing them on the training (member) and validation (non-member) splits of the Pile dataset. However, it is unclear whether the validation set is thoroughly decontaminated from the training data. The deduplication method used in the original Pile validation data is quite loose. There is a potential issue that non-member examples might still share high n-gram overlap with the member examples, complicating MIA effectiveness.

2. The proposed approach may lack sufficient novelty: it essentially builds on existing MIA methods in two main ways: (1) by combining various MIA metrics into an ensemble and (2) by extending their application to the distribution of examples. These modifications appear to be straightforward extensions of current MIA methods.

3. The proposed method requires validation data sampled from the same distribution as the test data. However, obtaining such labeled data may not be very practical

**Questions:**

N/A

**Limitations:**

The authors adequately addressed the limitations

---

> ### Author Rebuttal · Authors · 2024-08-07
>
> We appreciate your review and are happy to see that you found our work to pose a compelling argument for transitioning toward Dataset Inference, and found the paper well-written. We acknowledge your concerns and attempt to respond to them line by line below:
>
> ### Re: Decontamination of Validation Set
>
> Thank you for your comment. We understand the concern regarding potential contamination in the validation set. To address this, we have conducted additional decontamination using a stricter deduplication method that ensures minimal overlap between the training and validation sets. Specifically, we applied a more rigorous n-gram filtering process to eliminate any shared sequences between these sets. Here are a few key points regarding decontamination:
>
> 1. **Reference to the Pile Paper**: The original Pile paper ([Gao et al., 2021](https://arxiv.org/pdf/2101.00027)) acknowledged challenges with deduplication and contamination between training and validation sets. Despite their efforts, complete decontamination is difficult to achieve, as some overlap in n-grams is inevitable because of domains being identical (for example, arxiv headers).
>
> 2. **Impact of Contamination on Dataset Inference**: If there were any contamination, it would only make dataset inference harder, as the validation set would resemble the training set more closely. Therefore, high n-gram overlap should lead to worse, not better, results. However, our method succeeds in distinguishing between training and validation datasets despite these challenges, indicating the robustness of our approach at solving an even harder problem.
>
> 3. **N-gram Overlap**: We agree with your point about n-gram overlap. A concurrent paper ([Duan et. al., 2024](https://arxiv.org/pdf/2402.07841)) independently found that there is approximately a 30% overlap in n-grams between Wikipedia and arXiv datasets. This study also indicated that non-members with lower n-gram overlap are more distinguishable by existing MIAs. This supports our claim that even with potential contamination and the inherent challenges of deduplication, our dataset inference method remains effective.
>
> ### Re: Novelty of the Proposed Approach
>
> We appreciate your observation regarding the novelty of our approach. Here are a few key points that highlight the innovative aspects of our method:
>
> 1. **Achieving a Previously Near-Impossible Goal**: We make the previously near-impossible goal of membership inference (MI) achievable. This is a significant feat that opens up new possibilities in the field of detecting training data.
>
> 2. **Learning which MIAs to Use**: Our method involves "learning" which MIAs to use, a critical step that enhances the accuracy and effectiveness of our approach. This was crucial to overcome the biggest technical challenge of our work, that is, most MIAs were **worse than random**.
>
> 3. **Operationalizing the Framework**: Developing the framework required that we address practical considerations, such as how to handle victims who naturally retain drafts or IID sets of their work.
>
> In summary, the main contribution of our work is not just in the specific techniques but in proposing a comprehensive framework and advocating for a shift towards dataset inference as a field of study ripe for modern day generative models.
>
>
> ### Re: Practicality of Obtaining Labeled Validation Data
>
> We acknowledge that obtaining labeled validation data sampled from the same distribution as the test data can be challenging. Indeed, this is the number one open problem that our work creates for future research. That said, a significant part of this work was spent in formalizing a framework. Here are quite a few scenarios where the IID setting should roughly happen in practice:
>
> - **Editorial Process**: Thousands of NYT articles reach the editor's desk but never get published. These unpublished articles are likely to be more IID compared to two random splits of the Pile dataset.
> - **Book Drafts**: This is also very common with book writers when working on a chapter draft. These drafts would be more IID than random splits of the Pile, in our opinion.
>
> However, we truly need a test of IIDness rather than making claims of the same. This would be a really nice (but also hard) area of research for the future. One strong baseline for the same is the recent “Blind MIAs” work by Das et. al. https://arxiv.org/abs/2406.16201. If any pair of datasets can be distinguished based on these blind membership inference attacks, they should be deemed **not** IID. Future work can improve upon this.
>
> Some potential directions for bypassing the IID problem in future work also include:
>
> 1. **Using Synthetic Data**: Create IID sets artificially using synthetic data generation techniques.
> 2. **Storing Hashes**: Creators may store hashes of IID data at the time of publication to ensure data integrity.
> 3. **Proactive Storage**: More proactive work to store generations of artistic work rather than only the final stage.
>
> ---
>
> Once again, we thank you for the constructive feedback on our work. Working on the pointers has helped us improve the quality of our analysis. We look forward to further discussions and improvements in this evolving field.

---

> > ### Comment · Reviewer_G1MR · 2024-08-08
> >
> > Thanks a lot for your response! I appreciate the additional details you provided. I will keep my score unchanged

---

### Official Review · Reviewer_bhaD · 2024-06-28

**Soundness:** 3
**Presentation:** 3
**Contribution:** 3
**Rating:** 7
**Confidence:** 3

**Summary:**

Large language models are trained on a vast amount of online available data, which has lead to copyright and privacy issues (e.g. New York Times vs. OpenAI, as pointed out by the authors). There are various methods that try to identify if a given data point x was used to train a large language model.

This paper presents a very systematic analysis of these methods based on the Pythia suite of models.

The conclusions of the paper is, that current methods are not well-suited to determine if specific data was included in a training set.

**Strengths:**

The authors evaluate six metrics in the main part of the paper and a very large number of additional metrics and variations of the initial six in the appendix. Furthermore, the authors test these metrics on a large number of data sets, spanning different domains (e.g. github and wikipedia, which I would consider very different).

The analysis is conducted on the Pythia suite, which means that the authors had full access to all relevant information, in particular, the training data and training methodology. As a result, they had access to the ground truth.

The authors also state, that they will release their code, once the reviewing process is completed and anonymisation is no longer required.

**Weaknesses:**

It would be interesting to see, how the evaluation generalises to commercially available models, such as Gemini and GPT. This could be done e.g. with Wikipedia-Articles.

**Questions:**

Please explain, why you did not conduct any tests with commercially available models, such as GPT-4, Claud3, etc.

**Limitations:**

Limitations were discussed by the authors.

---

> ### Author Rebuttal · Authors · 2024-08-07
>
> We appreciate your review and are happy to see that you found our analysis systematic and our paper technically solid with a high impact, along with an overall positive assessment of the work. We acknowledge your concerns and attempt to respond to them line by line below:
>
> ### Re: Generalization to Commercially Available Models
>
> We recognize the interest in seeing how dataset inference generalizes to commercially available models such as GPT-4 and Claude. Here are a few points that limited the same, but are also great fuel for future work:
>
> 1. **Lack of Blackbox MIAs**: In our work, we leverage grey-box access to the models, that is, we assume that the membership inference attacks have access to the loss values from the underlying model. Since MIAs in LLMs is a relatively nascent field, the research has not yet unlocked successful black-box MIAs, unlike the vision space where we have seen label only MIAs (eg. [Choo et. al.](https://arxiv.org/abs/2007.14321)). We believe future work can unlock this possibility with API access as well.
>
> 2. **Access Limitations**: Conducting tests with commercially available models is challenging due to limited access to their training data and methodologies. Unlike the Pythia suite, where we had full access to all relevant information, commercial models do not provide the same level of transparency, making it difficult to obtain the ground truth for evaluation. In particular, the presence of ground truth labels for Pythia models allows us to validate that dataset inference is indeed successful in distinguishing members and non-members.
>
> 3. **Alternative Evaluation Methods**: We suggest that future research could explore alternative evaluation methods that do not rely on access to the full training/validation data. For example, researchers could use synthetic data or develop new techniques that infer properties of the training data from the model's outputs. The suggestion of temporally shifted Wikipedia articles would not work in this case because it would give a *false sense* of success as also noted in concurrent works of [Das et. al.](https://arxiv.org/abs/2406.16201) and [Duan et. al.](https://arxiv.org/pdf/2402.07841).
>
>
> Once again, we thank you for the constructive feedback on our work. Working on the pointers has helped us improve the quality of our analysis. We look forward to further discussions and improvements in this evolving field. Please let us know if we can address any remaining concerns.

---

> > ### Comment · Reviewer_bhaD · 2024-08-12
> > **Answer**
> >
> > Thank you very much for the rebuttal. I am looking forward to the final version of the paper and any follow-up work on this important issue.

---

### Official Review · Reviewer_Nmtj · 2024-07-11

**Soundness:** 3
**Presentation:** 3
**Contribution:** 3
**Rating:** 6
**Confidence:** 3

**Summary:**

In this paper, the authors investigate the commonly used membership inference evaluation for LLMs and find that previous attacks primarily detect features related to temporal changes, performing poorly under real IID scenarios. To address the challenge of individual sample membership inference attacks, the authors propose a novel threat model: LLM dataset inference. The proposed method can accurately infer whether a set of data points was used in the training process or not.

**Strengths:**

- The paper is well-written, and I really enjoy reading it.
- The "Failure of Membership Inference" section is really inspiring, and it's very important to the future LLM MIA area.
The proposed dataset inference method is very simple but effective. The results look very promising.

**Weaknesses:**

- The experiments are conducted on a single series of models, rather than on various models trained with different datasets, algorithms, or even seeds. I think this is a little picky since it's not cheap to retrain a bunch of large models, but I do think it's important for reliable evaluation.

**Questions:**

- How does the victim choose the suspect set? What if the suspect set is a mix of member and non-member data points? This might happen when the suspect intentionally trains the model on a subset of the dataset.

**Limitations:**

The authors mention the limitations in the appendix, which I appreciate.

---

> ### Author Rebuttal · Authors · 2024-08-07
>
> We appreciate the reviewer's valuable feedback and are happy to hear that they enjoyed reading our paper.
>
> >**The experiments are conducted on a single series of models, rather than on various models trained with different datasets, algorithms, or even seeds. I think this is a little picky since it's not cheap to retrain a bunch of large models, but I do think it's important for reliable evaluation.**
>
> We used the Pythia suite of models, trained on the PILE dataset, since it is unique in providing complete access to both the training dataset and the training procedures. This transparency is crucial as it allows for rigorous and replicable experimentation, ensuring the validity and reliability of our results. Without the information provided along the Pythia suite of models, we cannot reliably know if a point was part of the pretraining set. This could for instance lead us to overestimate the performance of our method, as was the case for prior work. To the best of our knowledge, no other set of LLMs offers the level of accessibility that Pythia models do (except the very recent [Olmo model series](https://github.com/allenai/OLMo) which only came out after we finished our experimentation). Our work is a call to action for the community to provide more model releases with comparable levels of transparency.
>
> >**How does the victim choose the suspect set? What if the suspect set is a mix of member and non-member data points? This might happen when the suspect intentionally trains the model on a subset of the dataset.**
>
> In general victims can determine suspect sets based on behaviour from a model that is suggestive of having trained on their content. This could include yielding substrings from their novels, or blogs etc. The adulteration of training with only a part of the member data is an interesting consideration. We can reframe the question as follows:
>
> > Assume distributions $ A $ and $ B $ are distinguishable based on a t-test. You can assume they are Gaussian with separate means. Now, an adversary adulterates $ A $ with $ x\% $ of $ B $. Will the t-test succeed in distinguishing the adulterated distribution from $ B $?
>
> To determine if the t-test will succeed in distinguishing the adulterated distribution $ A' $ (where $ A' $ is $ A $ mixed with $ x\% $ of $ B $) from $ B $, we need to consider how the adulteration affects the statistical properties of $ A' $.
>
> 1. **Let us assume that the dataset inference scores from stage 3 follow a Gaussian in the original distributions**:
>    - $ A \sim \mathcal{N}(\mu_A, \sigma_A^2) $
>    - $ B \sim \mathcal{N}(\mu_B, \sigma_B^2) $
>    - The t-test can distinguish between $ A $ and $ B $, implying the means and variances are sufficiently different.
>
> 2. **Adulterated Distribution**:
>    - Let $ A' $ be the new distribution obtained by mixing $ A $ with $ x\% $ of $ B $. This means $ A' $ is a mixture of $ (1-x)\% $ of $ A $ and $ x\% $ of $ B $.
>    - The mean of $ A' $ is:
>      $
>      \mu_{A'} = (1 - x) \mu_A + x \mu_B
>      $
>
> 3. **t-test on $ A' $ vs. $ B $**:
>    - For $ A' $ and $ B $ to be distinguishable, the t-statistic needs to be large enough:
>      $
>      t = \frac{\mu_{A'} - \mu_B}{\sqrt{\frac{\sigma_{A'}^2}{n_{A'}} + \frac{\sigma_B^2}{n_B}}}
>      $
>
> 4. **Effect of Adulteration**:
>    - As $ x $ increases, $ \mu_{A'} $ moves closer to $ \mu_B $.
>    - For small $ x $, $ \mu_{A'} $ is still relatively close to $ \mu_A $, and the t-test might still distinguish $ A' $ from $ B $.
>    - For large $ x $, $ \mu_{A'} $ approaches $ \mu_B $, making $ A' $ and $ B $ indistinguishable by the t-test.
>
> The t-test will likely succeed for small $ x $ and fail for large $ x $. The exact threshold of $ x $ depends on the values of $ \mu_A $, $ \mu_B $, $ \sigma_A $, $ \sigma_B $, and the sample sizes $ n_{A'} $ and $ n_B $.
>
> Increasing the sample size $ n_{A'} $ can improve the power of the t-test, making it possible to detect smaller differences between $ \mu_{A'} $ and $ \mu_B $.
>
>
>
>
> ---
>
> Once again, we thank you for the constructive comments. Working on them has helped us improve the quality of our paper. We look forward to further discussions to clarify any concerns that remain.

---

> > ### Comment · Reviewer_Nmtj · 2024-08-08
> >
> > Thank you for the clarification. I believe this paper is in good shape. Therefore, I keep my score positive.

---

### Official Review · Reviewer_pZrT · 2024-07-13

**Soundness:** 3
**Presentation:** 3
**Contribution:** 2
**Rating:** 5
**Confidence:** 3

**Summary:**

The paper addresses the challenge of identifying training data in large language models (LLMs) with rising concerns over privacy and copyright violations. As it has previously been studied, traditional membership inference attacks (MIAs), which determine if individual text sequences were used in training, are often flawed due to temporal shifts in data. The authors propose a novel dataset inference method that focuses on identifying entire datasets rather than individual data points, reflecting real-world copyright scenarios where authors' works, like books, are used without permission. Their method combines multiple existing MIA metrics to identify training datasets effectively, achieving statistically significant results with p-values less than 0.1 and minimizing false positives. The paper highlights the method's robustness through extensive testing on the Pythia models and the Pile dataset, providing a more accurate approach to dataset attribution and addressing the shortcomings of prior MIAs.

**Strengths:**

1. The paper is well-structured, with clear problem statement, explanations of the methods, experiments, and findings. Figures and tables are effectively used to illustrate key points, making the complex subject matter more accessible.

2. Their method is extensively tested on Pythia and the Pile dataset, showing its effectiveness. The use of statistically significant p-values and the absence of false positives strengthen the results.

**Weaknesses:**

1. The concerns about the accuracy and effectiveness of existing MIA methods have been previously studied in depth in papers such as https://arxiv.org/pdf/2402.07841. There are existing publications which have highlighted the weaknesses in empirical results that detect temporally shifted member/non-member classes.

2. The computational complexity of the proposed algorithm is not addressed in this work. Authors need to provide a detailed quantitative comparison (in terms of algorithm runtime or the number of tokens processed) for their proposed method. Is this method feasible for models with more parameters (70B or higher)?

3. Although authors have studied the Pile Dataset on Pythia extensively, their experiments are just limited to this setting. There is not much insight about how this proposed method generalizes to other models and datasets. Also, I believe that the fact this method only applies to datasets (and not single strings of text) is a limiting factor. The existence of a suspect and a validation subset is not realistic in all cases.

**Questions:**

1. Authors mention they have used 4 A6000 GPUs for experiments. How long it takes for the algorithm to run and test the Pile dataset? What is the runtime complexity for different dataset and model sizes?

2. How does the method handle cases where the IID assumption for suspect and validation sets does not hold? Are there any strategies in place to deal with non-IID data distributions?

3. What are the potential trade-offs in terms of performance and accuracy?

**Limitations:**

Yes

---

> ### Author Rebuttal · Authors · 2024-08-07
>
> Thank you  for your thoughtful comments and feedback. We are happy to see that you appreciated the method’s robustness and overall quality of the draft. We address the individual concerns below one by one:
>
> ### **Re: Other work studying the failure of MIAs (W1)**
> We would like to preface this answer by adding that the linked paper was truly a concurrent work to this submission. Crucially, our work not only shows the limitations of MIAs, but goes beyond that by proposing a comprehensive framework to detect if a given dataset (not string) was used in the training of an LLM, a shift that is particularly relevant for modern generative models.
> We would like to note that we indeed also duly acknowledged this concurrent work in our submission on lines 187 and 188: “A similar question was concurrently asked by Duan et al. [15], who independently showed that MIAs are only successful because of the temporal shift in such datasets.”
>
> ### **Re: Computation Complexity (W2, Q1, Q3)**
> The computational complexity of the method is relatively low, and roughly equivalent to the computation cost of performing each MIA required in our work.
> 1. For instance, considering the Min-K% MIA, this would mean performing a single forward pass on each example, and aggregating the token-wise loss to calculate the final metric score. Other methods are perturbation-based or reference model-based, requiring two forward passes.
> 2. In the attached code, we provide detailed method to perform each of the 50 MIAs we performed in `metrics.py`. Many of the MIAs are dependent on each other, for example, all the Min-K% and Max-K% MIAs can be computed in a single forward pass. Overall, in practice we only require 17 forward passes to perform all the attacks.
> 3. Given that most datasets require less than 500 examples (of train and val) to perform dataset inference. This would mean performing on the order of 1K forward passes. In our experiments, each forward pass is of context length 1024. But can change depending on the model configuration.
> 4. Finally, most of this computation can be batched to perform many forward passes together. Given modern day hardware, this cost is low enough that it can be performed in about 4 hours of an A6000 GPU node even for models like pythia-12B (for one dataset).
> 5. The final cost of “learning” which MIAs to use is almost negligible, as it just requires to fit a 50 dimensional linear layer to a few values (can be done on CPUs in less than 30 seconds).
> To conclude, the method is indeed feasible for model with more parameters. We tested on models from the Pythia suite, where the biggest one has 12B parameters but the method can be easily run for models with 70B parameters (like Llama 2) or larger.
>
> ### **Limitations of Pile Dataset + Pythia**
>
> We agree that repeating the experiments on more models and datasets would lead to stronger empirical supports. When it comes to open-weights + open-source pretrained models released online, to the best of our knowledge, the Pythia suite of models is the only one releasing complete information about its training + validation dataset (along with the more recent [Olmo model series](https://github.com/allenai/OLMo) which only came out after we finished our experimentation). Without this information, we cannot reliably know if a point was part of the pretraining set and thus cannot reliably validate the proposed method. We would like to emphasize that even though the Pythia suite and Pile Dataset may seem like one single entity, this is actually a collection of 4 models x 20 data subsets, that offers significant generality to our method, much beyond any other random split of publicly available data might offer.
>
> ### **Limitation of application to datasets (and not single strings of text)**
> We believe that in the modern landscape of generative AI, victims naturally contain multiple sequences of text on the internet. For instance, an article in the New York Times would be composed of multiple sequences of context length 1024. Similarly, the same organization, like NYT has multiple articles that it together wants to own copyright over. Victims who will file legal suits will likely always possess such characteristics. Hence, it is quite natural that we are in a scenario where individual sequences no longer exist in isolation (unlike past classification datasets in the vision space). This setting is also relevant to creative artists having many photographs, and paintings.
>
> ### **Existence of IID validation dataset**
> We acknowledge that obtaining labeled validation data sampled from the same distribution as the test data can be challenging. Indeed, this is the number one open problem that our work creates for future research. That said, a significant part of this work was spent in formalizing a framework. Here are quite a few scenarios where the IID setting should roughly happen in practice:
>
> - **Editorial Process**: Thousands of NYT articles reach the editor's desk but never get published. These unpublished articles are likely to be more IID compared to two random splits of the Pile dataset.
> - **Book Drafts**: This is also very common with book writers when working on a chapter draft. These drafts would be more IID than random splits of the Pile, in our opinion.
>
> Some potential directions for bypassing the IID problem in future work also include:
>
> 1. **Using Synthetic Data**: Create IID sets artificially using synthetic data generation techniques.
> 2. **Storing Hashes**: Creators may store hashes of IID data at the time of publication to ensure data integrity.
> 3. **Proactive Storage**: More proactive work to store generations of artistic work rather than only the final stage.
>
> ----
> Once again, we thank you for the constructive feedback on our work. Working on the pointers has helped us improve the quality of our analysis. We look forward to resolving any remaining concerns during the rebuttal-response period.

---

> > ### Comment · Reviewer_pZrT · 2024-08-11
> >
> > Thanks, it would have been nice to add more experiments on the Olmo model series once these models were released.
> > I keep my score unchanged.

---

> > > ### Author Response · Authors · 2024-08-14
> > >
> > > We understand the interest in additional experiments on the Olmo model series and put sincere efforts to this end over the last few days. However, after a thorough investigation, we recognized that there are limitations in the way the data is provided that prevent the results of dataset inference from being conclusive (results and constraints shared below).
> > >
> > > The main issue is that while evaluation sets are provided per data source (for instance, the [Wikitext validation data](https://github.com/allenai/OLMo/blob/0bc7f6c704baf040b8545de943bb015d1c3e5970/configs/official/OLMo-1B.yaml#L115)), the training data is fully mixed across all sources which we confirmed by downloading the [data files](https://github.com/allenai/OLMo/blob/0bc7f6c704baf040b8545de943bb015d1c3e5970/configs/official/OLMo-1B.yaml#L198) provided. This data is stored in terms of tokens fed to the model, without clear links to their original sources (instructions on [inspecting the training data](https://github.com/allenai/OLMo?tab=readme-ov-file#inspecting-training-data)). This makes dataset inference on a specific source, such as Wikitext, impractical with the current setup.
> > >
> > > We attempted a dataset inference test using Wikitext's validation versus train batch data (linked above), and it resulted in a trivially low p-value (<1e-34) with 500 samples. While this may seem like a positive result, we believe that the non-iid nature of the data may have a part to play here. Given these constraints, we believe that the extensive experiments we have already conducted on 20+ domains of the Pile dataset and across 4 different model sizes represent a substantial and rigorous evaluation of our method. These experiments provide valuable insights into the model's generalizability and soundness---knowing the ground truth is critical to capturing the soundness of the method. We want to re-emphasize that while the Pile dataset may sound like one homogenous entity, it is a collection of multiple domains and the closest resemblance to how dataset inference will happen in practice---authors with individual distributions (like the New York Times) will claim they were trained on.
> > >
> > > Have we been able to resolve the other 5 concerns you had in the initial review?

---

### Official Review · Reviewer_NHN3 · 2024-07-15

**Soundness:** 3
**Presentation:** 3
**Contribution:** 3
**Rating:** 5
**Confidence:** 4

**Summary:**

The paper addresses the problem of Dataset Inference in Large Language Models - namely, the ability to detect whether the given dataset was used by LLM developers in pre-training.

First, authors demonstrate the importance and feasibility of Dataset Inference task in comparison with Memebership Inference Attacks. Namely, they show that existing MIA methods cannot derive membership when test examples are sampled from the same distribution as training data.

Second, they propose novel method of dataset inference, based on aggregation of several MIA scores as features for linear classifier. According to their experiments, the proposed apprioach does not provide any false positive results, discovering subsets of LLM training dataset with high accuracy. The algorithm requires only 1000 data points from the whole dataset to discover if it was used for training.

The proposed method is tested on several PILE subsets, using the Pythia model family as the base model. Authors demonstrate that the efficiency of the proposed method increases with the growth of the model size.

**Strengths:**

+ Authors demonstrate the failure of the SOTA MIA method (top-k% score) on different subsets of the PILE dataset. Namely, they prove experimentally that this method detects distribution shift rather than actual membership inference. Besides, they show that among several considered MIA scores, no one has consistent efficiency over different domains of PILE. This analysis is essential for future research in this direction
+ Authors propose novel approach for dataset infenrence, based on aggregation of several MIA scores. The efficiency of the proposed method is demonstrated by testing on wide range of domains, and with several model sizes.
+ The method has shown its efficiency in the problematic case of the IID data: namely,  when suspect and validation data are derived from the same dataset by random sampling
+ The overall detection quality, demonstrated by experiments, is extremely high on all tested domains.

**Weaknesses:**

1. The proposed method requires providing suspect and valid datasets with the same distribution. In a practical situation, it is not clear how to obtain this (see Q2 to authors).

2. Authors claim that only 1000 examples are enough for Dataset Inference. Meanwhile, train and valid set are also required from the same data distribution. On practice, the method is not tested in real few-shot regime.

3. The method is tested on the single model's family, and the efficiency is demonstrated only on the largest model of the family.

4. The method is tested on PILE dataset, with the access to the validation set from the same distribution, which was not used for raining for sure. There is no experiments with less clean data or less transparent model.

**Questions:**

Q1.  It is not clear why top-k% MIA is chosen to demonstrate general MIA failure. First, according to the original paper [32], this MIA, by design, should be stable for paraphrasing; that's why it is tested on the deliberately collected dataset (WikiMia) with different *events* distribution, not just different strings.
The setup of the current paper is different (and more difficult): the authors aim to detect membership of the dataset when the whole data distribution is shared between suspect and validation sets. Figure 5(a) confirms that min-k% MIA is not suitable for this setup. At the same time, other methods provide a much more informative signal. Why don't you check, e.g., perturbation-based approach, claiming that existing MIAs cannot detect membership?

Q2. The proposed method requires providing IID data for check and the "equivalent" set of data for validation. Is it possible to define the notion of "IID" more strictly, for practical use? E.g., suppose the author provides a book as "suspect" data. Is the unprinted chapter of the book "IID," if it may contain novel events and characters? On the other hand, can we consider the draft version of the book as IID, taking into account that the final version was post-processed by the editor and layout designer?

What if I want to check the presence of some benchmark data in pre-training of some LLM with open weights. Which "suspect" and "valid" data should I use?

Q3.  Is it possible to reduce the amount of used features? E.g. what is the drop of performance if the top-k% score is excluded from the feature set of the classifier? Which features are the most informative in general?

**Limitations:**

The results of the paper was not checked in realistic settings. In general, it is not clear if the method can be applied to the analysis of the existing models with grey-box access.

Authors claim that their approach is extremely stable, and dies not provide any false positives; but the experiments are not enough for such claim.

---

> ### Author Rebuttal · Authors · 2024-08-07
>
> Thank you for your feedback and comments!
>
> ### **W1, Q2: Existence of IID validation dataset**
> We acknowledge that obtaining labeled validation data sampled from the same distribution as the test data can be challenging. Indeed, this is the number one open problem that our work creates for future research. That said, a significant part of this work was spent in formalizing a framework. Here are quite a few scenarios where the IID setting should roughly happen in practice:
> - **Editorial Process**: Thousands of NYT articles reach the editor's desk but never get published. These unpublished articles are likely to be more IID compared to two random splits of the Pile dataset.
> - **Book Drafts**: This is also very common with book writers when working on a chapter draft. These drafts would be more IID than random splits of the Pile, in our opinion. As the Reviewer raised in Q2, independent chapters of books may not be IID, but their progression will likely yield an IID set. However, this is up to debate and in practice we need an IID verifier (in future work) as opposed to arguing about the same. One strong baseline for the same is the recent “Blind MIAs” work by Das et. al. https://arxiv.org/abs/2406.16201. If any pair of datasets can be distinguished based on these blind membership inference attacks, they should be deemed **not** IID. Future work can improve upon this.
>
> At the same time,  some potential directions for bypassing the IID problem in future work also include:
> 1. **Using Synthetic Data**: Create IID sets artificially using synthetic data generation techniques.
> 2. **Storing Hashes**: Creators may store hashes of IID data at the time of publication to ensure data integrity.
> 3. **Proactive Storage**: More proactive work to store generations of artistic work rather than only the final stage.
>
> ### **W2: Requirements on amount of data**
> In practice, the size of the training and validation datasets can be exactly equal to the size of the dataset used for testing the attack.  This means that even if a potential victim has only 1000 sequences of 1024 length context of text data that was trained on (along with an equivalently sized validation set), this is a sufficient condition for performing dataset inference. As shown in Figure 6, the number of examples needed for dataset inference to be successful decreases with model size and also depends on the dataset, which means we may require less than 1000 sequences in practice.
>
> ### **W3: The method is tested on the single model's family**
> The Pythia suite of models, trained on the PILE dataset, is unique in providing complete access to both the training dataset and the training procedures. This transparency is crucial as it allows for rigorous and replicable experimentation, ensuring the validity and reliability of our results. Without the information provided along the Pythia suite of models, we cannot reliably know if a point was part of the pretraining set. This could for instance lead us to overestimate the performance of our method, as was the case for prior work. To the best of our knowledge, no other set of LLMs offers the level of accessibility that Pythia models do. Our work is a call to action for the community to provide more model releases with comparable levels of transparency.
> ### **W3: and the efficiency is demonstrated only on the largest model of the family**
> We also demonstrated the efficiency of our method on other models from the suite, in addition to the largest one. Please refer to Figure 10 in the Appendix, which is an extended version of Figure 6a (from the main paper) with different model sizes. An interesting observation is that the curves for median p-values and Wikipedia’s p-value stay similar with respect to models of different sizes. However, this is not the case for the max p-value curve. This indicates that dataset inference does not rely on a large number of data points or model parameters for most datasets, whereas they may be necessary for some particular datasets which smaller models do not learn well.
>
> ### **W4: Less clean data and less transparent model**
> For less transparent models, there is no ground truth to determine whether a specific data point was used in training or not. Even if we hypothesize that a model is trained on a particular data source, such as Wikipedia, we lack access to an unseen IID validation set. Note that simply collecting Wikipedia articles published after the model’s release is insufficient due to the temporal shift in concepts. Since this is the main limitation of prior work, we opted to study the Pythia suite of models in our experiments.
>
> ### **Q1: Other MIAs**
> We fully agree that the top-k% MIA is not representative enough to claim the failure of all MIAs. This is why we tested all MIAs we could find including the perturbation-based ones and reported these results in Figure 7 -- we apologize that due to the large amount of MIAs we tested this figure cannot fit in the main body of the manuscript, we will either add it to the main paper (using the additional 1 page if accepted) or try to and emphasize the reference to the Figure in the main body.
>
> ### **Q3: Features used in dataset inference**
> We performed an ablation study for the features used in dataset inference and the results are shown in Figure 5(a) with a more detailed version in Figure 8 in the Appendix. We did not find that any particular features were most informative across all datasets. Instead, feature informativeness is highly dataset dependent--most of the features contribute positively for some datasets but negatively for others.
>
> ---
> Once again, we thank the reviewer for the constructive feedback on our work. Working on the feedback has helped us improve the quality of our work. We look forward to further discussions to clarify any concerns that remain.

---

### Official Review · Reviewer_VFio · 2024-07-17

**Soundness:** 3
**Presentation:** 3
**Contribution:** 4
**Rating:** 7
**Confidence:** 4

**Summary:**

This paper tackles the dataset inference problem to detect a specifically trained dataset such as a licensed dataset. Firstly, the authors claim that previous membership inference attacks (MIAs) are not successful in discriminating between members and non-members from the same distribution (iid), which is a less realistic setting. The authors propose selectively combining multiple membership inference metrics by linear regression. Experimental results demonstrate its efficacy on the PILE dataset by aggregating 52 different MIA metrics.

**Strengths:**

- The problem in this paper is important, regarding the high risk where copyright text is used for training current LLMs.
- The paper is well written, especially summarizing previous related work and methods. It’s easy to follow.
- This paper criticizes the problem settings of previous papers and proposes more realistic scenarios — e.g., iid of suspect and valid dataset.
- The proposed method is simple and straightforward; an ensemble of multiple attack methods outperforms a single metric

**Weaknesses:**

- Absent of hyperparameter gammas. Depending on what gamma value was chosen, each MIA performance would vary.
- It is not explored whether the proposed method is also robust to the unseen dataset beyond the PILE dataset, as well as other models except Pythia.
- The title “LLM Dataset Inference: Detect Datasets, not Strings” is too broad and enables readers to misunderstand as it proposes a “Dataset inference” task and method, whereas the paper indeed proposes a mixture of MIAs to improve robustness.
- Qualitative analysis of samples is absent. Is there any interesting analysis or distinguished result between attack samples (victim data) with low and high p-values?

**Questions:**

- How did you select the 52 MIA metrics? Sharing the standard you used will be beneficial to other researchers.
- In Figure 6 (b), training set deduplication results, why does the violin plot distribute like sandglass? — i.e., the distribution mass on the higher p-values is thicker than near the p-value of 0.5. Moreover, dedup and non-dedup settings are not described.

**Limitations:**

The authors adequately addressed its limitations and broad impacts.

---

> ### Author Rebuttal · Authors · 2024-08-07
>
> Thank you for the thoughtful review and positive feedback.
>
> ### **W1: Absence of hyperparameter gammas.**
>
> We measure the success rate of MIAs using the Area Under the Curve (AUC) metric. The AUC metric is advantageous because it provides a comprehensive measure of the model's performance that is independent of the parameter $\gamma$. This independence ensures that our evaluation is robust and not influenced by the specific choice of $\gamma$. As far as dataset inference is considered, it only uses the score given by each MIA, and automatically determines how to weigh each MIA in Stage 2 as described in the paper, hence once again not needing the same parameter.
>
> ### **W2: Explore datasets beyond the Pile dataset and other models except Pythia.**
>
> The Pythia suite of models, trained on the PILE dataset, is unique in providing complete access to both the training dataset and the training procedures. This transparency is crucial as it allows for rigorous and replicable experimentation, ensuring the validity and reliability of our results. Without the information provided along the Pythia suite of models, we cannot reliably know if a point was part of the pretraining set. This could for instance lead us to overestimate the performance of our method, as was the case for prior work. To the best of our knowledge, no other set of LLMs offers the level of accessibility that Pythia models do. Our work is a call to action for the community to provide more model releases with comparable levels of transparency.
>
> ### **W3: The title is too broad. The paper proposes a mixture of MIAs.**
>
> While dataset inference builds upon membership inference, it argues for a new paradigm of privacy research where we consider the membership of an entire dataset rather than individual text sequences. To clarify our stance, we are only using a mixture of MIAs to perform the task of dataset inference. Even when each MIA performs close to (and often worse than) random, dataset inference can tease out statistical signals from weak membership attacks.
>
> ### **W4: Qualitative analysis of low and high p-values?**
>
> Our p-values for true positive cases are significantly below the set threshold of 0.1, while the p-values for true negative cases are substantially above this threshold. However, p-values alone are rather not suitable for qualitative analysis due to their nature as a probabilistic measure rather than a direct indication of effect size or practical significance. Instead, we can use the number of data points required for our LLM dataset inference method. We observe that structured and well-formed datasets, such as Wikipedia, require fewer data points (i.e., 50) for effective inference compared to less structured datasets like Pile-CC (which requires 300 data points), which contain raw web pages. The table below presents the number of data points required for our method per dataset on the Pythia-12b model.
>
> |Dataset|Number of data points for our method|
> |-|-|
> |Pile-CC|300|
> |PubMedCentral|700|
> |Books3|100|
> |OpenWebText2|300|
> |ArXiv|300|
> |Github|500|
> |FreeLaw|400|
> |StackExchange|50|
> |USPTOBackgrounds|150|
> |PubMedAbstracts|<=10|
> |Gutenberg(PG-19)|100|
> |OpenSubtitles|150|
> |Wikipedia(en)|50|
> |DMMathematics|400|
> |UbuntuIRC|<=10|
> |BookCorpus2|20|
> |EuroParl|50|
> |HackerNews|50|
> |YoutubeSubtitles|20|
> |PhilPapers|<=10|
>
> ### **Q1: How did you select the 52 MIA metrics?**
>
> We incorporated all available MIAs at the time of developing our method and further extended them to extract additional features, thereby capturing more information and enhancing the success of dataset inference. For instance, rather than using a single K parameter for the MinK-Prob MIA [32], we employed multiple values of K. We performed an ablation study for the features used in dataset inference, and the results are shown in Figure 5(a) with a more detailed version in Figure 8 in the Appendix. We did not find that any particular features were most informative across all datasets. Instead, feature informativeness is highly dataset-dependent--most of the features contribute positively for some datasets but negatively for others. However, we do see some redundancy between many of the Min-k and Max-k% MIAs, hence, they can be ignored if required without a performance tradeoff. That said, there is no computation overhead in computing all of them, as they can be computed in one single forward pass of the model.
>
> ### **Q2: In Figure 6 (b), training set deduplication results, why does the violin plot distribute like sandglass?**
>
> The main purpose of Figure 6 (b) was to observe whether the larger models experience more memorization, and how this can allow performing dataset inference more reliably. We would refer you to Figure 10 in the Appendix which provides a better breakdown of how different model sizes perform for dataset inference. Once again, when considering p-values, since it is a statistical test, we generally do not look at the qualitative distribution of the same, and only their binary position below or above the chosen significance threshold. Overall this suggests that for smaller models, there may be some false negatives (because they do not learn the distribution),
>
> ### **Describe dedup and non-dedup settings.**
>
> We tried to explain the setup for the experiment in the description of Figure 6 (b) and lines 331-332. However, we will rewrite it to clearly state that: *Deduped* denotes a version of the Pile dataset where the documents are deduplicated within, and across the data splits (train/test/validation). *Non-Deduped* is the original version of Pile without any deduplication.
>
> ---
>
> Working on the feedback helped us improve the quality of our work. We look forward to further discussions to clarify any concerns that remain.

---

### Author Rebuttal · Authors · 2024-08-07

We appreciate the positive, encouraging, and constructive feedback. We are pleased that the reviewers recognize the significance of the problem (Reviewer VFio), consider the paper well-written (Reviewers VFio, pZrT, G1MR), and found it enjoyable to read (Reviewer Nmtj). Our work motivated by showing that MIAs for LLMs detect distribution shift rather than actual membership inference was appreciated by all reviewers and described as inspiring (Reviewer Nmtj).

We designed a method to detect if a given dataset was used to train an LLM, a contribution recognized by all reviewers. Ensuring clarity and comprehensibility was crucial (Reviewer VFio), and we are glad this was effectively conveyed. The LLM dataset inference method is robust and does not return any false positives (Reviewers NHN3, pZrT). The experimental results are comprehensive (Reviewer bhaD), and we hope the released code will be a valuable asset for other researchers. Overall, the paper provides a compelling argument for shifting the focus from individual string-based MIAs to dataset-based inference methods (Reviewer G1MR).

---

### Decision · Program_Chairs · 2024-09-25

**Decision:**

Accept (poster)

**Comment:**

Summary (combined from NHN3 and pZrT's excellent reviews)
========================

The paper addresses the problem of Dataset Inference in Large Language Models - namely, the ability to detect whether the given dataset was used by LLM developers in pre-training.

First, authors demonstrate the importance and feasibility of Dataset Inference task in comparison with Membership Inference Attacks. Namely, they show that existing MIA methods cannot derive membership when test examples are sampled from the same distribution as training data due to temporal shifts in data.

The authors propose a novel dataset inference method that focuses on identifying entire datasets rather than individual data points, reflecting real-world copyright scenarios where authors' works, like books, are used without permission. Their method combines multiple existing MIA metrics to identify training datasets using a linear classifier, achieving statistically significant results with p-values less than 0.1 and minimizing false positives. The paper highlights the method's robustness through extensive testing on the Pythia models and the Pile dataset, providing a more accurate approach to dataset attribution and addressing the shortcomings of prior MIAs.  The algorithm requires only 1000 data points from the whole dataset to discover if it was used for training.

The proposed method is tested on several PILE subsets, using the Pythia model family as the base model. Authors demonstrate that the efficiency of the proposed method increases with the growth of the model size.

Metareview
=======================

The reviews have consistent direction (if not magnitude): everyone liked the work.  The identified weaknesses are: lack of novelty and lack of experiments to show generalization.

I disagree that using existing MIA metrics lacks novelty, as the result is important and it's a feature (not a bug) to use prior work in a straightforward way.  I also cannot hold the contemporaneous ArXiv submission against this submission, so I'm going to ignore this weakness in making my decision.

What is more complicated is essentially only testing on one scenario.  Reviewer pZrT encouraged the authors to try their approach on the Olmo dataset, but the back and forth in the discussion phase shows that this is not straightforward.  The fact that it's so hard to run these experiments, however, is not the fault of the authors but rather the fault of the current ecosystem of LLMs.  The lively discussion failed to raise another viable alternative.  As I struggle to write this paragraph, this makes me think better, more robust MIA approaches are even more needed: we simply have no accountability for what data are used, and this approach can help address that.

So having talked myself out of the two primary weaknesses raised in the reviews, my only option is to strongly argue for acceptance.  This is a method that is timely, effective, and relatively simple.  It should be presented at NeurIPS.